# Hydration and Mechanical Properties of Cement Kiln Dust-Blended Cement Composite

**DOI:** 10.3390/ma17194841

**Published:** 2024-09-30

**Authors:** Woo-Seok Lee, Young-Cheol Choi

**Affiliations:** Department of Civil and Environmental Engineering, Gachon University, Seongnam 13120, Republic of Korea; dntjr8071@gachon.ac.kr

**Keywords:** cement kiln dust, hydration, compressive strength

## Abstract

This study aims to investigate the effects of cement kiln dust (CKD) on the hydration reactions and mechanical properties of cement, and to evaluate its potential for use as a supplementary cementitious material (SCM). The key variables are the CKD type and the replacement ratio. Cement paste and mortar specimens containing CKD were prepared to examine their effects on the cement hydration and mechanical properties. The effect on hydration was assessed using setting time measurements, heat of hydration tests, and thermogravimetric analyses (TG). In addition, compressive strength tests were conducted to evaluate the effect of CKD on the mechanical properties of the cement. The results indicated that CKD promoted early-age cement hydration and enhanced the early-age mechanical properties. However, owing to its lack of pozzolanic reactivity, it did not significantly affect long-term hydration. Given that the effects of CKD vary slightly depending on its chemical composition, careful consideration of CKD’s properties suggests that its potential use as an SCM is promising.

## 1. Introduction

Addressing global warming has emerged as a critical global priority, leading to extensive research and development of various technological strategies aimed at reducing greenhouse gas emissions. The cement industry is a major contributor to both energy consumption and greenhouse gas emissions, accounting for approximately 7% of the global CO_2_ emissions. Consequently, significant efforts have been directed towards mitigating its environmental impact [1,2]. Among the various strategies, the reduction in clinker content through substitution with supplementary cementitious materials (SCMs) has been identified as one of the most effective approaches [3]. Typical SCMs, such as fly ash (FA), ground granulated blast furnace slag (GGBS), and silica fume (SF), enhance the hydration process and improve the physical properties of cement [4,5]. However, the availability and types of SCMs are largely dependent on shifts in related industries such as steel production and power generation, resulting in an increasing demand for alternative SCMs. Notably in the United States, the transition of power plants has significantly reduced the availability of traditional supplementary cementitious materials (SCMs), such as fly ash [6].

In response to this demand, the cement industry has investigated the utilization of cement kiln dust (CKD), a by-product generated during the cement manufacturing process, as a potential SCM. CKD consists primarily of fine particles collected during kiln operations and typically contains oxides such as CaO, MgO, Fe_2_O_3_, Al_2_O_3_, and SiO_2_; its composition varies depending on the production process and raw materials used [7]. Despite its high specific surface area and fine particle size, which suggest its potential as an SCM, CKD is frequently disposed of in landfills or recycled back into the cement production process, owing to the lack of effective utilization methods [8,9]. Further research is required to develop efficient strategies for the utilization of CKD as an SCM, thereby contributing to a reduction in CO_2_ emissions in the cement industry [10].

Al-Harthy et al. investigated the effects of CKD on the compressive strength of concrete [11]. Their study involved preparing concrete specimens with CKD replacement levels of up to 30% and measuring the compressive strength at various curing ages. The findings revealed a general decline in the compressive strength as the CKD replacement ratio increased, particularly at high water-to-binder ratios. However, when CKD was used as partial replacement, at levels of up to 5%, the compressive strength was not adversely affected. Ramakrishnan explored the properties of concrete incorporating CKD [12]. Their study found that a 5% replacement of CKD resulted in a slight delay in setting time; however, other properties of fresh concrete remained largely comparable to those of conventional concrete. Bhatty investigated the combined use of various CKD types with fly ash (Class C and Class F) and ground granulated blast furnace slag (GGBS) [13,14,15]. They reported that the use of CKD alone led to reductions in setting time, compressive strength, and workability.

According to studies by Maslehuddin et al. [7] and El-Aleem et al. [16], replacing approximately 10% of the cement by weight with CKD significantly reduced both the initial and final setting times by approximately half. The researchers attributed this effect to the high lime and alkali contents present in CKD. Ramakrishnan and Balaguru investigated the durability of concrete with 5% CKD replacement [17]. Their findings indicated that at this replacement level, CKD did not significantly affect the resistance of concrete to freeze-thaw cycles. Konsta-Gdoutos et al. investigated the potential use of CKD as a new cementitious binder [18]. They examined the effectiveness of four types of CKD as GGBS activators. They reported that CKD, owing to its high alkali and sulfate contents, served as an excellent activator for pozzolanic materials. In addition, CKD enhanced the activation of GGBS, promoting the formation and accumulation of calcium silicate hydrate (C-S-H), which in turn improved the compressive strength. Building on these previous studies, recent research has investigated various methods of utilizing CKD as an SCM. Current efforts are focused on understanding how different CKD replacement ratios and types of blended materials influence the physical and chemical properties of concrete. However, despite these ongoing investigations, there remains a need for more comprehensive studies on the effects of different CKD types and replacement ratios on cement hydration and mechanical properties.

Research utilizing CKD as an SCM has predominantly focused on evaluating the mechanical properties of mortar and concrete based on the CKD replacement ratio. However, comprehensive studies examining the influence of the chemical properties of CKD on cement hydration reactions and mechanical characteristics are lacking. This study aims to fill this gap by analyzing the effects of two types of CKD, each with distinct chemical compositions, on cement hydration reactions and mechanical properties. To this end, cement pastes and mortar specimens were prepared, with CKD type and replacement ratio as the primary variables. To assess the impact of CKD on cement hydration, scanning electron microscopy (SEM), setting time measurements, heat of hydration tests, and thermogravimetric (TG) analyses were conducted. Additionally, compressive strength was measured at various curing ages to evaluate the influence of CKD on the mechanical properties of cement.

## 2. Materials and Methods

### 2.1. Materials

In this study, research cement (RC) without mineral admixtures was used to investigate the hydration and mechanical properties of cement composites containing CKD. RC was produced by blending 95% clinker with 5% anhydrite. Two types of CKD have been identified in South Korea. CKD, generated during the cement manufacturing process, is collected from the volatile matter and dust particles carried by the exhaust gases of the rotary kiln. These particles are captured by electrostatic precipitators installed downstream of the cooling tower and heat exchanger. CKD typically constitutes 7–15% of the raw meal input and is characterized by its fine particle size. Currently, most CKD is recycled back into the cement manufacturing process without further utilization. This study employed two types of CKD: CKD (CA) from the ordinary Portland cement manufacturing process of Company A in South Korea, and CKD (CU) from the white cement manufacturing process of Company U in South Korea. Figure 1 presents optical images of the raw materials used in this study, showing that RC and CA exhibited a dark gray color, whereas CU appeared bright white. This color difference is likely due to variations in the composition of the clinker, which result from differences in the raw meal composition and manufacturing process. Both CA and CU partially agglomerated, owing to their high moisture content, and the powder surfaces appeared moist.

Table 1 presents the chemical oxide compositions of the raw materials as determined using an X-ray fluorescence (XRF) spectrometer. The density and Blaine fineness of the RC are 3.16 g/cm^3^ and 3601 cm^2^/g, respectively. The major components of CA are CaO, SiO_2_, Al_2_O_3_, and Fe_2_O_3_. The density and Blaine fineness of CA are 2.77 g/cm^3^ and 6830 cm^2^/g, respectively, indicating that CA consists of significantly finer particles compared to RC. The primary components of CU, like those of CA, are CaO and SiO_2_. However, owing to the differences in the raw meal and manufacturing process, CU has a lower Fe_2_O_3_ content and higher Al_2_O_3_ content than CA. The density of CU is 2.74 g/cm^3^, and its Blaine fineness is slightly higher than CA’s, at 7034 cm^2^/g, indicating very fine particles. Additionally, although CA contains almost no free CaO, CU has a higher free CaO content of 2.46%. The SO_3_ content of CU is slightly lower than that of RC. The free CaO content was determined using the ethylene glycol method in accordance with ASTM C 25 [19].

The LOI values for CA and CU were significantly higher than those for RC, which can be attributed to the differences in their compositions. Figure 2 shows the results of the TG analysis of the raw materials. As illustrated in the graph of Figure 2, a sharp weight change was observed around 700–800 °C, which corresponds to the decarbonation reaction of CaCO_3_ [20]. The CaCO_3_ contents of CA and CU, as shown in Figure 2, were 52.2% and 41.2%, respectively.

To further investigate the compositional and mineralogical characteristics of the RC and CKDs, an X-ray diffraction (XRD) analysis was performed, and the resulting XRD patterns are shown in Figure 3. The primary mineral phases in RC were identified as alite (C_3_S), belite (C_2_S), aluminate (C_3_A), ferrite (C_4_AF), and gypsum (CaSO_4_·2H_2_O). The dominant minerals for CA were calcite (CaCO_3_) and quartz (SiO_2_), whereas CU primarily consisted of calcite (CaCO_3_), lime (free CaO), quartz (SiO_2_), and anhydrite (CaSO_4_).

Figure 4 presents the SEM images of RC, CA, and CU. As shown in Figure 4, the particles of CA and CU exhibited irregular angular shapes, similar to those of RC, rather than spherical forms. However, the particle sizes of CA and CU were significantly smaller than those of RC. Typically, CKD is collected during the pre-calcination stage of clinker production using bag filters, resulting in a finer particulate state than that of RC. In addition, the particles may appear as aggregates or clusters, which could affect the dispersion of CKD within the mixture. Figure 5 illustrates the particle size distributions of RC, CA, and CU. The average particle size of RC was 17.0 μm, while those of CA and CU were 7.3 μm and 7.6 μm, respectively.

### 2.2. Mixture Proportions

Paste and mortar specimens incorporating CKD were prepared to investigate its effects on the hydration and mechanical properties of cement. The mix proportions used for specimen preparation are listed in Table 2. The primary variables were the type of CKD and the CKD replacement ratio by weight of the RC (ranging from 0% to 35%). ISO standard sand was used as the fine aggregate, and paste specimens were prepared using the same proportions as those listed in Table 2, excluding the fine aggregate.

### 2.3. Test Methods

X-ray diffraction (XRD) was conducted to examine the mineral composition of the raw materials. The XRD diffractometer used was the X’Pert Pro MPD (Empyrean by Malvern Panalytical), operating at 40 kV and 40 mA with CuKα radiation (λ = 1.54 Å). The analysis was performed from 10° to 65° in a continuous scanning mode at 0.01° intervals and a step time of 1 s per interval. The setting time of the cement paste containing CKD was measured according to ISO 9597, and the effect of CKD on the workability of the cement paste was evaluated using a mini-slump cone test in accordance with ASTM C 143. After preparing the CKD-containing cement paste, it was placed into a 50 mm diameter cone and slowly lifted. The diameter of the paste was measured once the flow stopped. The diameters were recorded with an accuracy of 1 mm. The compressive strength of the mortar containing CKD was measured after 3, 7, 28, 56, and 91 days of curing in accordance with ISO 679. Compressive strength tests were performed on the six specimens at each curing age and the average value was used as the result. The hydration heat of the CKD-containing cement paste was measured for 72 h using an I-CAL 2000HPC isothermal calorimeter (Calmetrix, Boston, MA, USA). One day prior to the test, the RC, two types of CKD, and distilled water were stored in a temperature- and humidity-controlled chamber at 23 °C, consistent with the equipment’s experimental conditions. After mixing the cement paste according to the proportions specified in Table 2, the mixture was placed in a plastic container for measurement and the results were calculated by considering the weight of the binder. To investigate the effect of CKD on the hydration reaction of the cement paste, TG analysis was performed at different curing ages (7, 28, and 91 d). TG analysis was conducted using an SDT Q600 (TA Instruments, New Castle, DE, USA), with the sample weight changes being automatically recorded as the temperature was increased up to 1000 °C in a nitrogen atmosphere.

## 3. Discussion

### 3.1. Workability and Setting Time

The results of the mini-slump cone test for the cement paste with varying amounts of CA and CU are presented on the left axis of Figure 6, whereas the amount of superplasticizer (SP) required to maintain the same flow value as Plain is indicated on the right axis. In this study, a polycarboxylate-based SP with a solids content of 20% was used. The Plain exhibited a slump of approximately 150 mm, and the amount of SP was adjusted to ensure that the flow of the paste containing CA and CU was maintained at 150 ± 5 mm. As the amounts of CA and CU increased, the flow rate of the cement paste decreased. This reduction in flow is likely due to the significantly smaller particle size and irregular angular shape of CA and CU compared to RC [11]. The paste containing CU exhibited a greater slump reduction than that containing CA. The higher free CaO content in CU compared to CA is believed to have contributed to the decrease in fluidity with increasing CU content owing to enhanced early-age reactivity.

Figure 7 illustrates the penetration depth over time of the Vicat needle in the cement paste containing CA and CU. As the CA and CU contents increased, the penetration depth curves shifted to the left, indicating a faster setting process. This trend is particularly pronounced in the cement paste containing CU. This behavior is consistent with the flow test results shown in Figure 5 and is likely attributable to the chemical composition of the CU.

Figure 8 depicts the initial and final setting times derived from the penetration depth measurements of the Vicat needle shown in Figure 7. The initial setting time for CA05 increased by approximately 2.7% compared to that of Plain. However, when the CA content exceeded 10%, a linear decrease in the initial setting time was observed. Specifically, the initial setting time for CA35 reduced by 13.6% compared to that of Plain. In addition, the final setting time consistently decreased with increasing CA content. For the cement paste containing CU, a trend similar to that observed with CA was noted, although the reduction in setting time was more pronounced. In CU05, the initial setting time increased by approximately 3.4% compared to that of Plain. However, when the CU content exceeded 10%, the initial setting time decreased more significantly than with CA, with CU35 showing a reduction of approximately 22.3% compared to Plain. The final setting time of the CU-containing cement paste also decreased as the CU content increased, with a more substantial reduction than the initial setting time. This trend was more pronounced in the CU-containing cement paste, similar to the results of the mini-slump test, and was likely attributable to the chemical composition of the CKD [21].

### 3.2. Hydration Properties

Figure 9 shows the heat flow and cumulative heat measurements as functions of CKD replacement ratios. Figure 9a shows the heat flow corresponding to different CA replacement ratios, revealing that as the CA content increased, the magnitude of the second peak decreased. The second peak of Plain was at 2.98 mW/g, whereas that of the CA35 showed a 16.8% reduction to 2.48 mW/g. This decrease in the magnitude of the second peak with CA replacement appears to result from the reduced proportion of cement, leading to a lower heat release during the hydration reaction [22]. Figure 9b illustrates the cumulative heat release results for the specimens containing CA. All data are expressed in milliwatts per gram of the binder. The cumulative heat release results indicated that all specimens with CA replacement exhibited a reduction in cumulative heat compared with Plain. The 72 h cumulative heat release for CA05, CA10, CA15, CA20, CA25, CA30, and CA35 decreased by 1.5%, 2.8%, 6.0%, 7.2%, 11.1%, 13.7%, and 17.9%, respectively, compared to Plain, following an almost linear trend. This reduction is attributed to the replacement of cement by CKD, which decreases the amounts of C_3_S, C_3_A, and C_2_S, the prime components responsible for heat release during the hydration reaction [23]. C_3_S, C_3_A, and C_2_S, the major components of cement, play critical roles in heat release during the early and late stages of hydration [24]. C_3_S and C_3_A contributed significantly to heat release in the early stages of hydration, but their proportions decreased with the replacement of CKD, resulting in a reduced early heat release. C_2_S, which contributes to the heat release during the later stages of hydration, also decreased in proportion to the replacement of CKD, leading to a reduction in the overall cumulative heat release [24]. CKD generally exhibits a lower reactivity than cement, which further diminishes the overall hydration reaction and cumulative heat release.

As shown in Figure 9c, similar to the specimens containing CA, increasing the CU content resulted in a decrease in the magnitude of the second peak. Specifically, the second peak for CU35 was measured at 2.30 mW/g, which was lower than that of CA35. Figure 9d illustrates the cumulative heat release results for the CU specimens. The 72 h cumulative heat release for the CU-containing samples (CU05, CU10, CU15, CU20, CU25, CU30, and CU35) decreased by 2.9%, 6.1%, 9.7%, 9.8%, 13.0%, 16.7%, and 19.2%, respectively, compared to Plain. This reduction was more pronounced than that observed for CA. The 72 h cumulative heat release for CU35 was only approximately 80% of that of Plain.

Figure 10 presents the 72 h cumulative heat of cement paste containing CA and CU. As shown in Figure 10, the cumulative heat of the cement paste containing CU was lower than that of the paste containing CA. CU contained a higher amount of free CaO than CA. Generally, free CaO reacts with water to form Ca(OH)_2_, a process which suppresses the heat release [25]. Consequently, the CU led to the formation of more Ca(OH)_2_ during the early stages of hydration, which enhanced the microstructure and further accelerated the hydration process. However, this also reduced the overall amount of heat released [25]. Additionally, CU has finer particles than CA, which increased the reactive surface area during hydration, thereby accelerating the initial hydration reaction. Nevertheless, this increased reactivity can contribute to a decrease in the total heat released [26].

### 3.3. Compressive Strength

Figure 11 shows the compressive strength of the mortar at different curing ages as a function of the CKD replacement ratio. As shown in Figure 11a, the results after seven days of curing indicate that when CA is substituted at levels up to approximately 10%, the compressive strength exceeds that of Plain. This early strength enhancement is likely due to the fine particle size of CA, with a fineness of 6830 cm^2^/g, which is significantly finer than that of RC. During the initial hydration reaction, fine CA particles fill the spaces between the cement particles, contributing to the filler effect and providing nucleation sites that accelerate the hydration process, thereby improving the early strength [27].

Previous studies have shown that CKD can positively affect the compressive strength of cement mortar at early ages. For instance, Alnahhal et al. reported that a small addition of CKD to cement mortar mixtures can accelerate the hydration process, leading to improved early strength [28]. The high level of fineness of CKD allows its particles to occupy the spaces between the cement particles, providing more reactive surfaces, increasing the initial rate of hydration, and filling the pores to enhance density. Densification of the pore structure improves the mechanical properties of the mortar. However, when the CKD replacement ratio exceeded a certain level, the compressive strength tended to decrease significantly [29]. This decline is attributed to a reduction in the content of key components such as C_3_S and C_2_S, which play a crucial role in the development of compressive strength, as CKD partially replaces cement [30]. Al-Harthy et al. reported that replacing cement with CKD reduces the content of strength-contributing components in mortar, resulting in a lower compressive strength compared to the plain concrete [11]. These findings are consistent with the results of this study, suggesting that although CKD can enhance early strength, excessive replacement has a detrimental effect. After early-age hydration, CA05 exhibited a compressive strength similar to that of Plain. However, when the CA content exceeded 10%, the compressive strength at 28 d and beyond was lower than that of Plain. While CA acts as a filler and positively influences the early hydration reaction, an increase in CA content over time results in a relative decrease in the proportion of key cementitious components such as C_3_S and C_2_S, leading to a reduction in strength development.

Figure 11b shows the compressive strength of the mortar at different curing ages as a function of the CU replacement ratio. Unlike CA, the replacement of CU resulted in a higher compressive strength at seven days of curing when CU is substituted at levels up to 20%, surpassing that of Plain. Similar to CA, this improvement in the early strength can be attributed to the filler effect resulting from the high fineness of CU (7034 cm^2^/g). The compressive strength at 7 days was even more enhanced with CU than with CA, likely because of the presence of free CaO in CU. The free CaO in CU reacts with water, accelerating the early hydration reactions and improving the microstructure of the cement paste through the formation of Ca(OH)_2_ [31]. However, when CU is incorporated beyond a certain percentage, a decrease in compressive strength is observed. This reduction is likely due to the decrease in the cement clinker components and the high content of free CaO [32]. After 28 d of curing, the difference in compressive strength between the samples containing CU and Plain became increasingly pronounced as the CU content increased. Notably, when CU was substituted at levels greater than 25%, a significant increase in the compressive strength difference was observed. By analyzing the slopes of the compressive strength curves beyond Day 7 in Figure 10 and Figure 11, it is evident that the slopes for the CA- and CU-containing specimens are nearly identical to or slightly steeper than those of Plain. This suggests that unlike fly ash (FA), the CKD used in this study, does not exhibit pozzolanic reactivity. Consequently, although the replacement of CKD enhances the early strength, it may be less effective in supporting long-term strength development.

### 3.4. TG Analysis

To examine the effect of CKD on cement hydration, thermogravimetric (TG) analyses were performed on the paste samples at 7, 28, and 91 d of age. Figure 12 shows the TG analysis results for the specimens incorporating CA. As shown in Figure 12, weight loss increased with the curing age for all specimens, and as the CA content increased, weight loss tended to decrease in the 50 °C to 200 °C temperature range. Typically, for cementitious materials, the weight loss in this temperature range is attributed to the dehydration of C-S-H and the decomposition of ettringite. These results are consistent with the heat flow data shown in Figure 9. In the 400 °C to 500 °C temperature range, weight loss is due to the decomposition of Ca(OH)_2_, and as hydration progresses with curing age, all specimens exhibited an increasing trend in weight loss in this temperature range. Similar to the behavior observed in the 50 °C to 200 °C range, the weight loss between 400 °C and 500 °C decreased as the CA content increased. Figure 13 shows the TG results for the specimens incorporating CU. As shown in Figure 13, the thermogravimetric analysis results for the CU specimens exhibited trends similar to those observed for the CA specimens. Table 3 summarizes the weight loss in the temperature ranges of 50–200 °C and 400–500 °C.

Figure 14 illustrates the Ca(OH)_2_ content according to the curing age for CA and CU specimens. The decomposition of H_2_O from Ca(OH)_2_ occurs within the temperature range of 400 °C to 500 °C, and can be obtained from TG results using the following equation
(1)$${\mathrm{Ca}(\mathrm{OH})}_{2}=\left(m_{400}-m_{500}\right)\times \frac{M_{{\mathrm{Ca}\left(\mathrm{OH}\right)}_{2}}}{M_{H_{2}O}}$$
where, $m_{400}-m_{500}$ is the mass loss between 400 °C and 500 °C, and $M_{{Ca\left(OH\right)}_{2}}$ and $M_{H_{2}O}$ are molar masses of Ca(OH)_2_ (74.09 g/mol) and H_2_O (18 g/mol), respectively.

As shown in Figure 14a, the Ca(OH)_2_ content in Plain was 12.88%, which increased with the curing age. At 91 d, the Ca(OH)_2_ content in Plain was approximately 1.4 times higher than that at 7 d. As the CA replacement ratio increased, the Ca(OH)_2_ content tended to decrease for all curing ages. Although the Ca(OH)_2_ content in the CA-incorporated specimens also increased with curing age, the rate of increase diminished as the CA content increased. For the CA30 specimen, the Ca(OH)_2_ content increased from 9.44% at 7 d to 12.34% at 91 d, which is 56% of the increase observed in Plain. This reduction is attributed to the replacement of CKD, leading to a decrease in the C_3_S and C_2_S components of the cement clinker, resulting in a proportional reduction in the hydration products, including C-S-H and Ca(OH)_2_. Figure 14b shows the results for the specimens incorporating CU, which exhibit a similar trend to that observed in the CA-incorporated cement paste.

## 4. Conclusions

This study investigated the effects of CKD on the hydration and mechanical properties of cement to evaluate its potential use as a supplementary cementitious material (SCM). The results indicate that CKD, owing to its high fineness, promotes early cement hydration and enhances early compressive strength. This suggests that CKD can be effectively used as an SCM in cement or concrete if an appropriate replacement ratio is applied. The key findings of this study are summarized as follows:

The CKD accelerated early cement hydration, leading to shorter setting times. The initial setting time of CA35 was reduced by 13.6% compared with that of Plain. Due to the influence of the lime it contains, CU resulted in an even shorter setting time than that of CA.As the replacement ratios of CA and CU increased, the peak magnitude of heat flow decreased. Additionally, the cumulative heat release decreased proportionally with the CKD content. This was attributed to the replacement of cement with CKD, which reduced the amounts of C_3_S, C_3_A, and C_2_S, which are the phases responsible for heat generation during hydration.The specimens incorporating up to 10% CA exhibited higher compressive strengths than the plain mixture. For specimens with 5% CA, the compressive strength was similar to that of Plain after 28 d. However, incorporating more than 10% CA resulted in a lower 28-day compressive strength than that of Plain, with a more significant reduction as the replacement ratio increased.CU showed a greater improvement in early compressive strength than CA. Specimens incorporating up to 20% CU exhibited a higher 7-day compressive strength than Plain. This enhancement is likely due to the higher free CaO and SO_3_ contents in the CU, which accelerated cement hydration. However, incorporating more than 25% CU led to a decrease in the compressive strength compared to Plain, with a more pronounced reduction in strength beyond 28 d.When comparing the slope of the compressive strength gain after 7 d, both CA and CU had smaller slopes than Plain, indicating that the CKD used in this study, unlike traditional fly ash, did not exhibit pozzolanic reactivity. This finding is consistent with the thermogravimetric analysis results, which showed that the Ca(OH)_2_ content of the specimens incorporating CKD did not decrease over time, further confirming the lack of pozzolanic activity.CKD has a chemical composition similar to that of limestone powder, and its impact on cement performance appears to be comparable. Moreover, given the slight variation in the effects of CKD, depending on its chemical composition, careful consideration of the specific characteristics of CKD suggests its high potential for use as an SCM.In order to utilize CKD as a SCM in concrete, it is necessary to quantitatively investigate the effects of CKD on the pore structure of concrete. Furthermore, extensive future research is required to assess its impact on durability, including carbonation resistance and chloride ion diffusion resistance.

## Figures and Tables

**Figure 1 materials-17-04841-f001:**
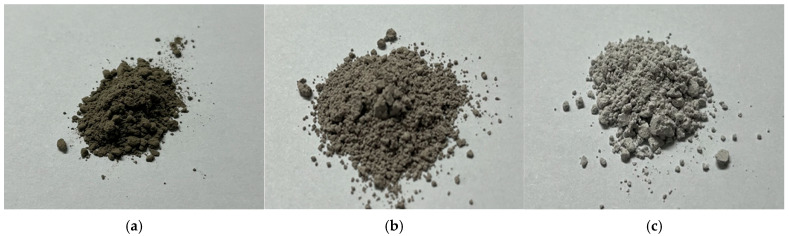
Optical images of raw materials used: (**a**) RC; (**b**) CA; (**c**) CU.

**Figure 2 materials-17-04841-f002:**
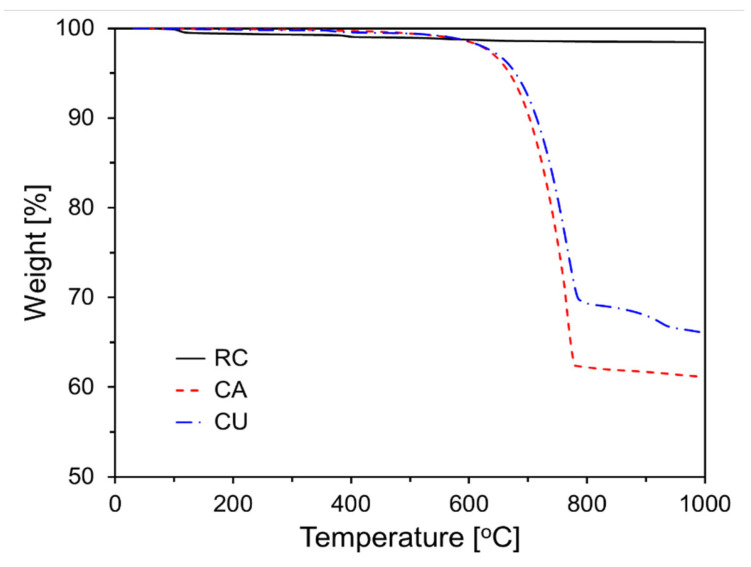
Thermogravimetric analysis of the raw materials.

**Figure 3 materials-17-04841-f003:**
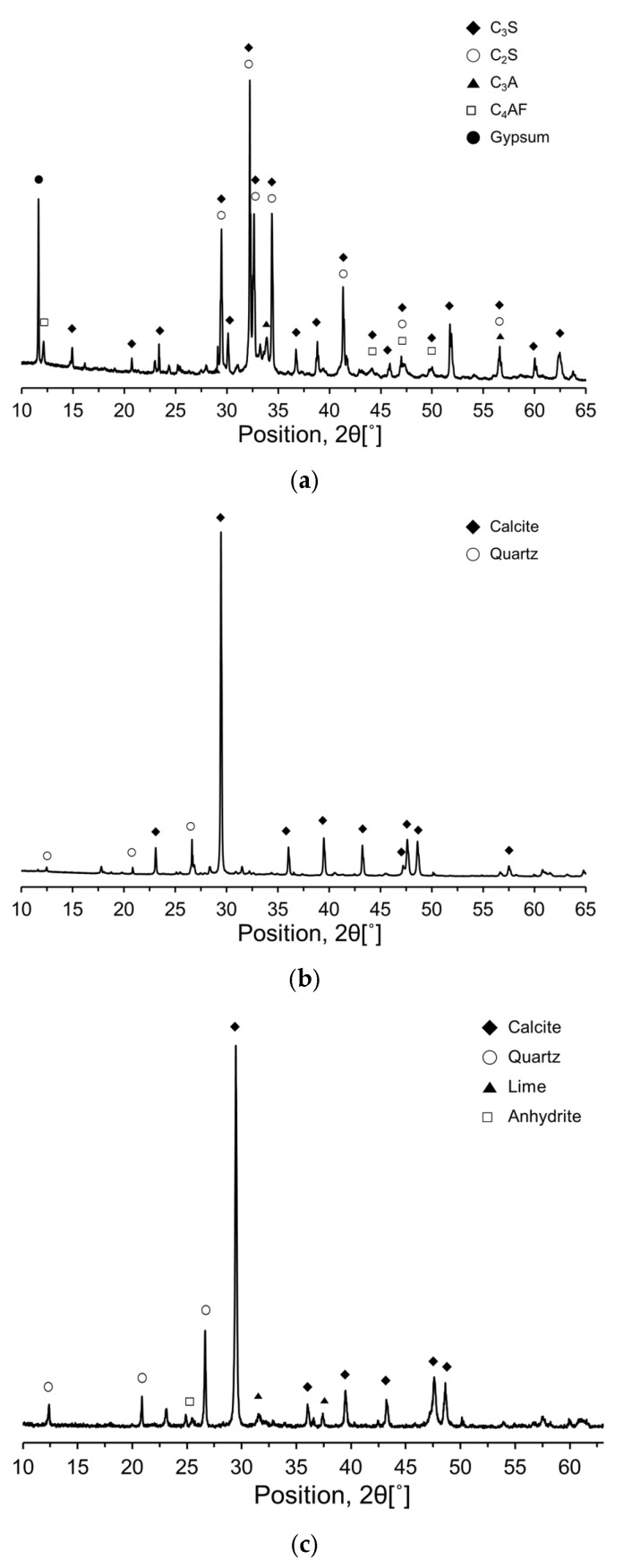
XRD patterns of the raw materials used: (**a**) RC; (**b**) CA; (**c**) CU.

**Figure 4 materials-17-04841-f004:**
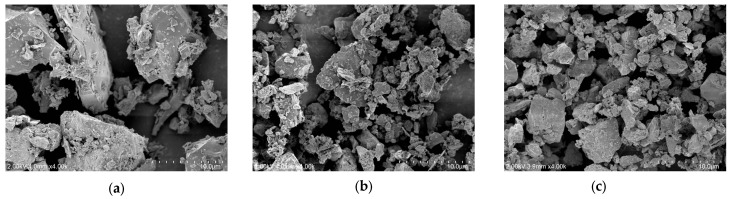
SEM images of the raw materials used: (**a**) RC; (**b**) CA; (**c**) CU.

**Figure 5 materials-17-04841-f005:**
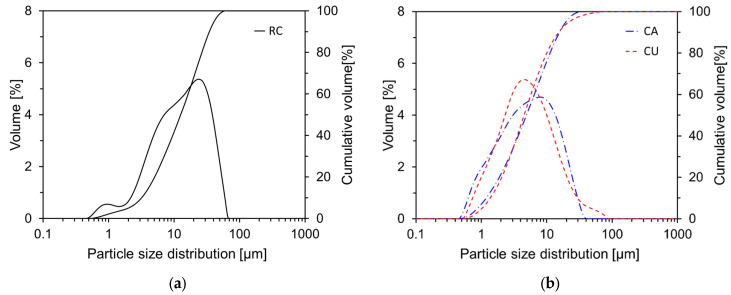
Particle size distributions of the raw materials used: (**a**) RC; (**b**) CA and CU.

**Figure 6 materials-17-04841-f006:**
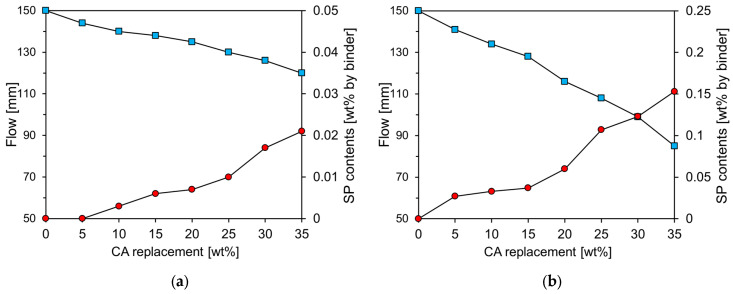
Flow test results of specimens containing CA and CU: (**a**) CA; (**b**) CU.

**Figure 7 materials-17-04841-f007:**
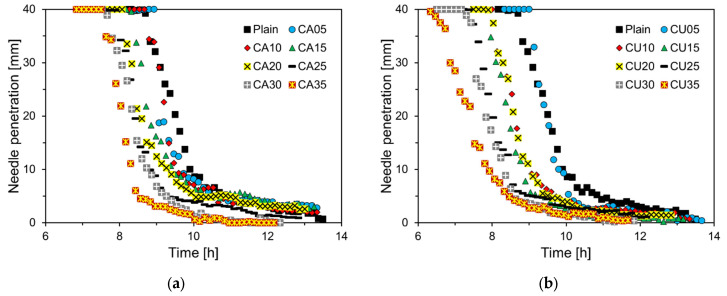
Vicat needle penetration depth results of specimens: (**a**) CA; (**b**) CU.

**Figure 8 materials-17-04841-f008:**
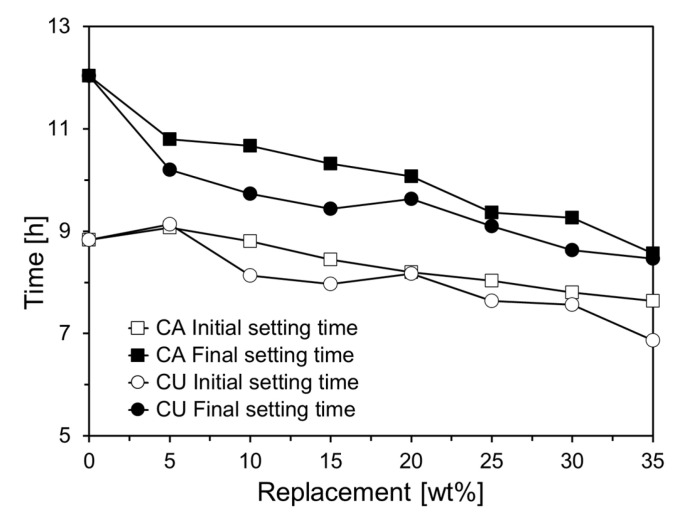
Setting times of specimens containing CA and CU.

**Figure 9 materials-17-04841-f009:**
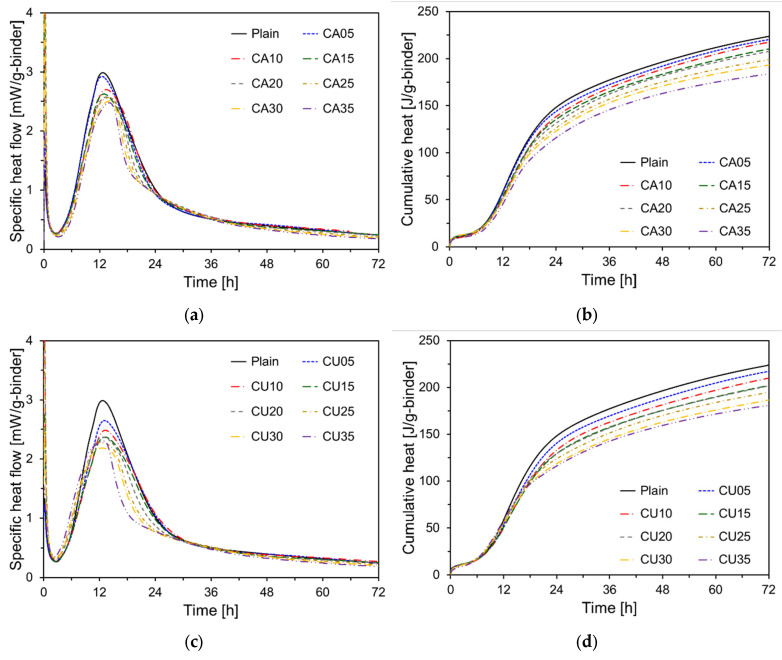
Results of the heat of hydration test for the paste containing CKD: (**a**) heat flow (CA); (**b**) cumulative heat (CA); (**c**) heat flow (CU); (**d**) cumulative heat (CU).

**Figure 10 materials-17-04841-f010:**
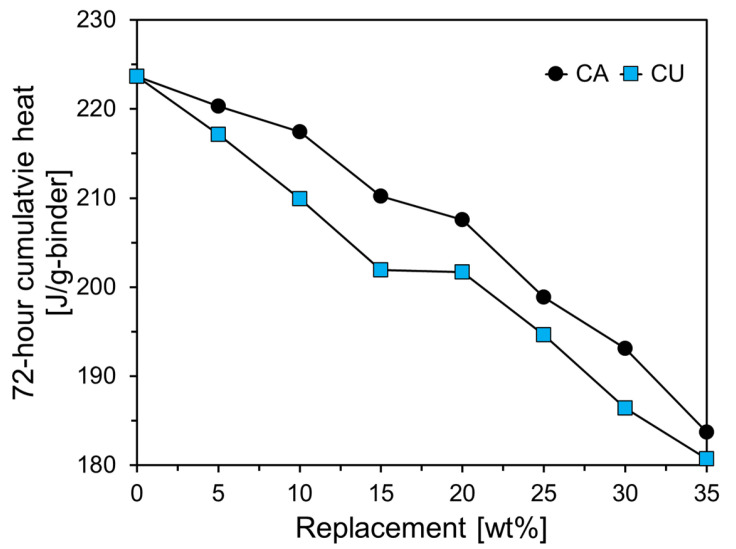
Comparison of 72 h cumulative heat results for cement paste containing CA and CU.

**Figure 11 materials-17-04841-f011:**
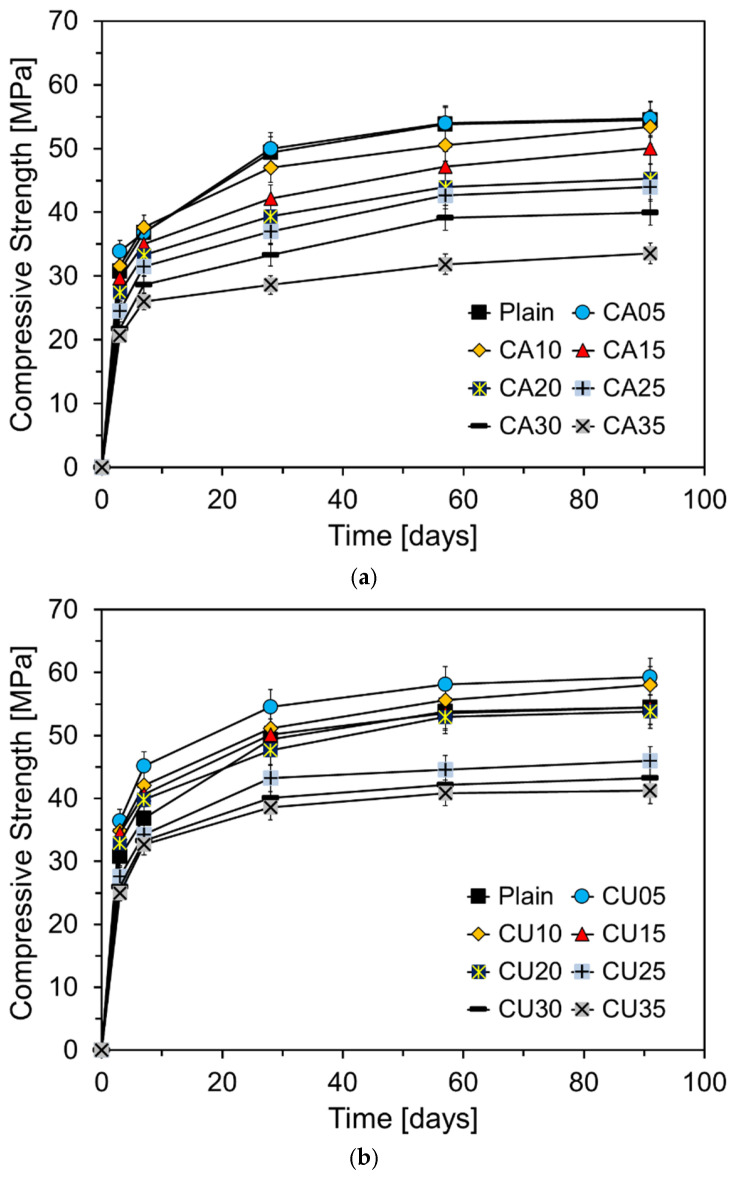
Compressive strength results of specimens containing CKD: (**a**) CA; (**b**) CU.

**Figure 12 materials-17-04841-f012:**
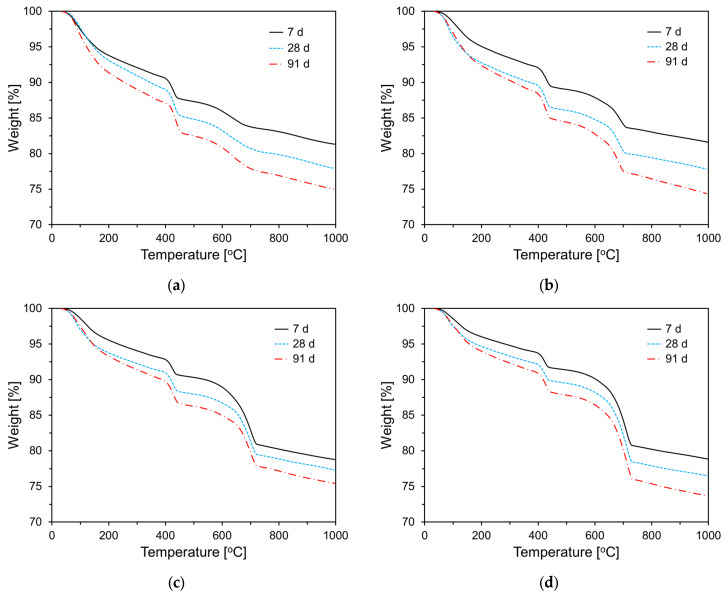
TG analysis results of specimens containing CA: (**a**) Plain; (**b**) CA10; (**c**) CA20; (**d**) CA30.

**Figure 13 materials-17-04841-f013:**
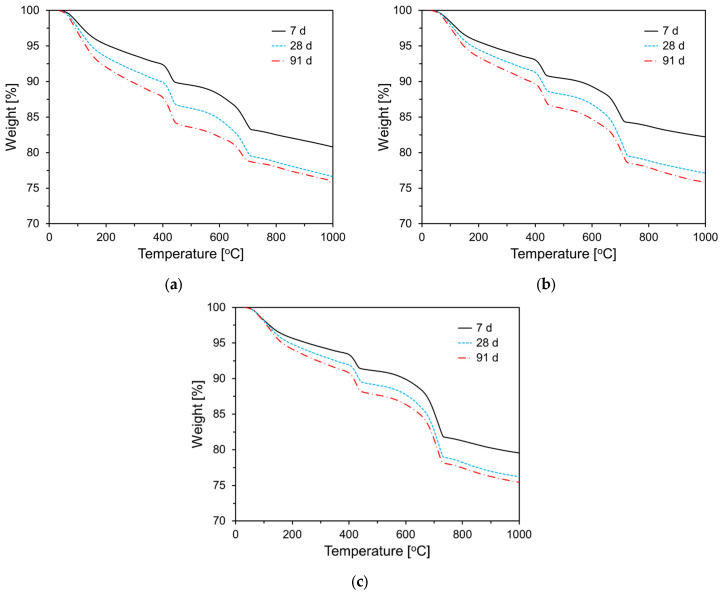
TG analysis results of specimens containing CU: (**a**) CU10; (**b**) CA20; (**c**) CA30.

**Figure 14 materials-17-04841-f014:**
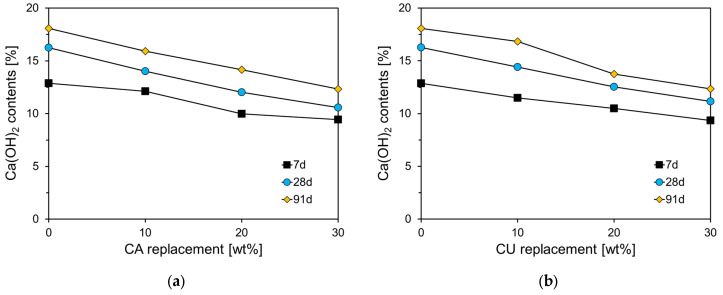
Ca(OH)_2_ contents of specimens containing CKD (**a**) CA; (**b**) CU.

**Table 1 materials-17-04841-t001:** Chemical oxide compositions of the raw materials used.

Chemical Composition	RC (%)	CA (%)	CU (%)
SiO_2_	19.20	11.46	20.78
Al_2_O_3_	4.31	4.69	5.82
Fe_2_O_3_	4.16	3.24	0.35
CaO	64.31	43.60	40.21
Free CaO	-	-	2.46
MgO	1.76	1.77	0.82
SO_3_	3.83	1.25	3.10
K_2_O	1.38	2.82	0.55
Na_2_O	-	-	0.21
LOI	0.38	37.06	29.97

**Table 2 materials-17-04841-t002:** Mixture proportions of the specimens.

Specimens	Water (g)	RC (g)	CA (g)	CU (g)	Sand (g)
Plain	50	100	-	-	300
CA05	50	95	5	-	300
CA10	50	90	10	-	300
CA15	50	85	15	-	300
CA20	50	80	20	-	300
CA25	50	75	25	-	300
CA30	50	70	30	-	300
CA35	50	65	35	-	300
CU05	50	95	-	5	300
CU10	50	90	-	10	300
CU15	50	85	-	15	300
CU20	50	80	-	20	300
CU25	50	75	-	25	300
CU30	50	70	-	30	300
CU35	50	65	-	35	300

**Table 3 materials-17-04841-t003:** Summary of weight loss in temperature ranges (50–200 °C, 400–500 °C).

Specimens	$m_{50}-m_{200}$ (%)			$m_{400}-m_{500}$ (%)		
	7 d	28 d	91 d	7 d	28 d	91 d
Plain	5.99	6.76	8.42	3.13	3.95	4.39
CA10	4.83	6.97	7.37	2.94	3.41	3.87
CA20	4.36	5.99	6.51	2.43	2.92	3.44
CA30	3.88	5.12	5.82	2.29	2.57	3.00
CU10	4.69	6.37	7.71	2.79	3.50	4.09
CU20	4.27	5.39	6.37	2.55	3.05	3.34
CU30	4.20	5.05	5.74	2.27	2.71	3.00

## Data Availability

The original contributions presented in this study are included in the article, further inquiries can be directed to the corresponding author.

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
