# Peer review of "Hydration and Mechanical Properties of Cement Kiln Dust-Blended Cement Composite"

_materials, 2024, doi:10.3390/ma17194841_

Round 1

Reviewer 1 Report

Comments and Suggestions for Authors

The paper is very well written, and the experimental results are also well explained. The manuscript was an interesting read and is suitable for publication. Some of the suggestions are included below for authors to consider.

1.      Line 44: If the CKD is recycled back into the production process, then is it utilization as SCM needed?

2.      Line 271: It would be good to show a comparison of the heat of hydration for CU vs CA through graph.

3.      What was the reason of reduced HoH for mixes containing CA considering their flow was also lower (and there seems to be no free CaO) ? Please explain this.

4.      Line 289 – If it is accelerating the hydration process, then shouldn’t that be visible in heat of hydration data? Why HoH for CA mixes are lower?

5.      Please include standard deviation in the compressive strength graphs  - also, it would be good to show all the mixes in a single graph for better comparison.

6.      It would be good to quantify the mass loss in a table for clearer comparison  - it is hard to draw comparison from the TG graphs. If the authors can show DTG, that would also be better for the analysis.

7.      Line 362: How did the authors calculate CH content ? How its’ decrease is related to the strength?

8.      Line 372 – Please correct the typo in the CH formula.

Reviewer 2 Report

Comments and Suggestions for Authors

Minor Revision

The manuscript has a sufficient workload and analyzes the effects of two different cement kiln dusts (CKDs) on the working and hydration properties of cement mortar. The results indicate that CKDs have the potential to be developed as supplementary cementitious materials (SCMs). However, the manuscript can be improved in the following aspects:

1. The arguments explaining how CKD (CA, CU) can improve the microstructure and mechanical properties of cementitious systems are currently divided into three parts: the filling effect of fine particles leading to increased compressive strength, the reduction of cement clinker leading to decreased hydration exothermic rate and amount, and the formation of calcium hydroxide inhibiting heat release. These explanations are rather superficial. It is recommended to provide additional in-depth arguments to make the conclusions more convincing. For example, further elaborating on the changes in the pore structure, hydration product formation, and their influence on the overall performance of the cementitious system would strengthen the discussion.

2.  In the thermogravimetric (TG) analysis section, the author has listed the decomposition of various hydration products of cement at different temperatures. However, there is a lack of specific conclusions on how the changes in hydration products affect the hydration process of the cementitious system after incorporating CKD. The analysis content should be expanded to directly address the statement "To examine the effect of CKD on cement hydration, thermogravimetric (TG) analysis was performed..." For instance, discussing the changes in the relative amounts of hydration products (e.g., calcium silicate hydrate, ettringite, portlandite) and their implications on the hydration kinetics and microstructure development would enhance the discussion.

3. The article layout can be further standardized:

- Combine the text of similar figures (e.g., hydration exothermic curve, compressive strength) to make the results of each group of specimens clearer and more concise.

- In the raw materials section, since many properties have already been listed in the tables, the need for individual descriptions can be reduced to avoid redundancy.

Reviewer 3 Report

Comments and Suggestions for Authors

In this study, the authors investigated the use of cement kiln dust as supplementary cementitious materials. Generally speaking, this is an interesting paper. The experiments are well designed, and the conclusions are supported by the results. Below are my comments.

1. There are some English problems. For example in abstract, incorporation rate means replacement ratio?

2. The authors mentioned the key variables are the CKD type. Is there a specific classification (type) for CKD? The materials sourced from different regions can always have different results.

3. The decreasing supply of traditional SCMs (such as fly ash) is due to the transition of power plants. In the US, we can already see the decreasing production of fly ash. I suggest adding relevant references to this point. (e.g., calcium nitrate effectively mitigates alkali-silica reaction by surface passivation of reactive aggregate.)

4. One of the main reasons for adding SCMs in concrete is to enhance the durability (such as the resistance to alkali-silica reaction, chloride transport etc.) by refining the pore structures and reducing total porosity. I suggest adding discussion on the future work regarding these aspects.

Round 2

Reviewer 1 Report

Comments and Suggestions for Authors

The authors have addressed the comments well. Paper can be accepted. 

Reviewer 3 Report

Comments and Suggestions for Authors

This paper has been revised and should be ready for publication.